# Association between Urban Built Environments and Moderate-to-Vigorous Physical Activity of Adolescents: A Cross-Lagged Study in Shanghai, China

**DOI:** 10.3390/ijerph19158938

**Published:** 2022-07-22

**Authors:** Zhengmao Guo, Yatao Xu, Shouming Li, Changzhu Qi

**Affiliations:** 1Postdoctoral Mobile Station of Physical Education, Wuhan Sports University, Wuhan 430079, China; qichangzhu@whsu.edu.cn; 2School of Physical Education, Shanghai Normal University, Shanghai 200234, China; 1000513268@smail.shnu.edu.cn; 3College of Physical Education and Health, East China Normal University, Shanghai 200241, China; 4Department of Physical Education & Health, Nanjing University of Finance & Economics, Nanjing 210046, China; xytao921@163.com

**Keywords:** built environment, moderate-to-vigorous physical activity, cross-lagged, adolescents

## Abstract

The aim of this study was to determine the relationship between the built environment and moderate-to-vigorous physical activity (MVPA) among adolescents aged 14–16 years. This study used a cross-lagged panel analysis to investigate the relationship between the urban built environment and adolescents’ MVPA in Shanghai, China. A total of 517 adolescents (275 boys and 242 girls) aged 14–17 years were recruited in Shanghai, China. Geographic information system technology was used to collect data on the built environment variables of the residential areas assessed. ActiGraphGT3X+ was used to monitor the physical activity of the adolescents at two time points (T1 and T2) spanning 2 years. The correlations between the T1 and T2 built environment variables were significant (*r* = 0.54–0.65, *p* < 0.05), and the T2 built environment was significantly better than the T1 built environment. The correlation between the T1 and T2 MVPA was significant (*r* = 0.28–0.56, *p* < 0.05), and the T2 weekend MVPA was higher than the T1 weekend MVPA. The T1 built environment could not predict the T2 weekday MVPA (*β* = 0.17, *p* > 0.05), but it positively predicted the T2 weekend MVPA (*β* = 0.24, *p* < 0.05). In conclusion, the urban built environment significantly affected weekend MVPA among adolescents.

## 1. Introduction

Physical activity levels in children and adolescents have decreased not only in developed countries, but also in low- and middle-income countries [1]. Environmental factors are important influences on changing levels of physical activity [2]. Research has shown significant associations of built environment factors such as traffic conditions, access to fitness facilities, and walking index with physical activity among children and adolescents [3,4]. Wong et al. found that the distance between home and school is one of the most important factors affecting children’s physical activity [5]. In addition, specific scenarios also affected physical activity among children and adolescents; for example, Mota et al. found that the number of fitness facilities near residences was positively correlated with physical activity among children and adolescents during their non-school hours [6]. Having safe areas where outdoor activities can be performed can attract children and adolescents to participate in outdoor activities [7]. Many results showed that there is a significant relationship between the urban built environment and physical activity among children and adolescents [8,9,10].

Previous studies have used a cross-sectional design and correlation analysis to investigate the relationship between the built environment and adolescents’ physical activity, but fewer studies have used a longitudinal design [10,11]. Cross-sectional comparative study data are easy to obtain and can describe the associations of different built environment variables with physical activity, but these data cannot determine the causal relationship between the built environment and physical activity [12]. A longitudinal design can be used to investigate the impact of changes in the built environment on physical activity by collecting data on changes over time in the built environment [13]. It can analyze the mechanisms by which the built environment variables affect physical activity and provide intuitive evidence of causality between changes in the built environment and physical activity.

This study aimed to use a cross-lagged panel analysis to investigate the relationship between the urban built environment and adolescents’ physical activity on weekdays and weekends and to reveal a causal relationship based on a longitudinal design.

## 2. Methods

### 2.1. Data Collection

All of the subjects aged 14–16 years old were randomly selected from 20 schools in the downtown area of Shanghai to participate in this study. The longitudinal investigation was conducted over two years. The first time was in September 2017 (T1), when 581 adolescents were investigated. The second time was in September 2019 (T2), and 517 adolescents were followed-up (39 adolescents changed schools and moved and the collected data were invalid for a further 25 cases). In total, 517 adolescents were assessed at school from 2017 to 2019.

There were no significant differences between the final effective sample and the missing T1 cases in gender (χ^2^_gender_ (1) =2.01, *p* = 0.10), built environment (t = 0.73, *p* = 0.52), or medium- and high-intensity physical activity (t = 1.8, *p* = 0.13), which showed that there was no obvious bias in the missing cases.

The study was approved by the Institutional Review Board of East China Normal University. Written informed consent was obtained from the adolescents involved in the study and their parents or guardians.

### 2.2. Variables

#### 2.2.1. Built Environment

The built environment variables investigated included nine variables in three categories: density, diversity design, and accessibility [14,15,16]. The density category involved two variables: population density and building density. The diversity design category involved three variables: street connectivity, per capita road length, and mixed land utilization rate. The accessibility category involved four variables: number of traffic stations, distance to traffic station, distance to fitness facility, and distance to commercial area.

Geographic information system (GIS; ArcGIS 10.2, Environmental Systems Research Institute, Redlands, CA, USA) technology was used to collect the built environment variables. The full-element digital map and vector map of downtown streets of Shanghai were imported into ArcGIS 10.2 to obtain the spatial data of the built environment. The spatial data included various fitness and leisure facilities, public places, the road network, the rail transit network, etc. Based on the radius of public facilities (such as bus stops and fitness trails) in Shanghai and the division of urban public sports space proposed by Cai et al. [17], the buffer radius was set at 1500 m. A buffer zone with a range of 1500 m was generated with the participants’ residential area as the center. Next, clip processing was performed on the buffer zone to extract the data on the built environment variables.

#### 2.2.2. Physical Activity

The “time-use epidemiology” research method can use “time allocation” as an entry point to study the physical activity of adolescents [18]. This method helped us categorize adolescents’ physical activity as physical activity on weekends and physical activity on weekdays, which allowed the relationship between the built environment and adolescents’ physical activity to be analyzed more accurately. The physical activity variables investigated were weekday and weekend moderate-to-vigorous physical activity (MVPA) duration.

ActiGraph GT3X^+^ (ActiGraph, Pensacola, FL, USA) was used to monitor the physical activity among adolescents who wore it continuously for 7 days. Before the test, all of the monitors were initialized and the participants were trained to wear, remove, and operate the monitor. Every day, it was confirmed that each participant wore their monitor at the correct times (7:00–21:00, except for bathing, swimming, and sleeping) and position (on the right hip). ActiLife 6 software (ActiGraph, Pensacola, FL, USA) was used to download the original data. The ActiGraph GT3X^+^ parameters are shown in Table 1.

### 2.3. Statistical Analysis

Analysis of variance (ANOVA) was used to assess the differences in the built environment and the adolescents’ physical activity with time (from T1 to T2) and gender. The relationship between the built environment and adolescents’ physical activity was examined. Cross-lagged panel modeling of the built environment and adolescents’ physical activity was conducted. The analysis was performed using SPSS 22.0 (International Business Machines Corporation, Armonk, NY, USA) and Amos 24.0 (International Business Machines Corporation, Armonk, NY, USA). The level of significance was set at *p* < 0.05.

The cross-lagged panel model is considered the most sensitive method for testing causality in longitudinally correlated data [19]. As shown in Figure 1, we assume any two variables are X and Y respectively, and the form of the cross-lag path analysis model is:
$$\begin{array}{ll} X_{t}= & \alpha_{1}+\beta_{1\times 1}+\rho_{1}Y_{1}+e_{1}\\ Y_{t}= & \alpha_{2}+\beta_{2}Y_{1}+\rho_{2}X_{1}+e_{2}(\alpha_{1},\alpha_{2}\;\text{are}\;\text{model intercepts};e_{1},e_{2}\;\text{are model residuals}) \end{array}$$


The path coefficient *ρ* is the basis for cross-lag path analysis to infer the causal time series relationship, which is the effect of the baseline measurement of the independent variable (such as X_1_) on the dependent variable (Y_t_) after controlling for the baseline state of the dependent variable (such as Y_1_). By comparing the path coefficients *ρ*_1_ and *ρ*_2_, the causal temporal relationship between variables can be determined, including the following four cases: if *ρ*_1_ is not significant and *ρ*_2_ is not significant, then there is no causal temporal relationship between X and Y; if *ρ*_1_ is significant and *ρ*_2_ is not significant, then the two variables are the one-way causal temporal relationship of Y → X; if *ρ*_1_ is not significant and *ρ*_2_ is significant, then the two variables are the one-way causal temporal relationship of X → Y; if both *ρ*_1_ and *ρ*_2_ are significant, then X and Y are a two-way causal temporal relationship regulating each other; in addition, if *ρ*_1_ > *ρ*_2_ and the difference between the two is significant, it can be further determined that the main causal sequential effect is Y → X, and vice versa, X → Y [20].

## 3. Results

The physical activity variables investigated were weekday and weekend MVPA. The built environment variables were divided into three categories: density, diversity design, and accessibility. Table 2 shows that the mean T2 density, diversity design, and physical activity variables were all higher than the corresponding T1 variables, while the mean T2 accessibility variables except for the number of traffic stations were lower than the corresponding T1 variables.

Table 3 shows that the main effects of time (T1 to T2) on population density (*p* = 0.040), building density (*p* = 0.034), street connectivity (*p* = 0.021), per capita road length (*p* = 0.047), mixed land utilization rate (*p* = 0.031), number of traffic stations (*p* = 0.024), distance to traffic station (*p* = 0.046), distance to fitness facility (*p* = 0.036), and distance to commercial area (*p* = 0.021) were all significant. The T2 built environment was significantly better than the T1 built environment (Table 2). The main effects of gender and the time × gender interaction effects on these built environment variables were not significant.

Table 4 shows that the main effect of time (from T1 to T2) on weekend MVPA was significant (*p* = 0.037), but its main effect on weekday MVPA was not (*p* = 0.254). The main effects of gender (*p* = 0.077; *p* = 0.473) and the time × gender interaction effects on these two physical activity variables were not significant (*p* = 0.514; *p* = 0.493).

The correlations between the T1 and T2 built environment variables were 0.54–0.65, and the correlations between the T1 and T2 physical activity variables (weekday and weekend MVPA) were 0.28–0.56; all of the correlations were significant (*p* < 0.05). In T1, the correlations of weekday or weekend MVPA with built environment variables were 0.34–0.55 (*p* < 0.05). In T2, the corresponding correlations were 0.29–0.65 (*p* < 0.05). The cross-time stability and synchronization correlation of the relationship between variables made the variables suitable for further cross-lagged panel analysis.

To further clarify the relationship between the built environment at T1 and T2 and adolescents’ weekday and weekend MVPA at T1 and T2, two cross-lagged panel models were established. The maximum likelihood method was used to test the model fit, gender was controlled for, and the residuals of different variables were allowed to be correlated at the same time. The model fit indices regarding the model of the built environment and weekday MVPA were good (χ^2^/df = 2.23, RMSEA = 0.07, SRMR = 0.04, GFI = 0.95, NFI = 0.89, CFI = 0.91, IFI = 0.92). The model fit indices regarding the model of the built environment and weekend MVPA were also good (χ^2^/df = 2.07, RMSEA = 0.06, SRMR = 0.03, GFI = 0.93, NFI = 0.90, CFI = 0.94, IFI = 0.91).

Figure 2 shows that the T1 built environment did not significantly predict the T2 weekday MVPA (*β* = 0.17, *p* > 0.05), and the T1 weekday MVPA did not significantly predict the T2 built environment (*β* = 0.08, *p* > 0.05). Figure 3 shows that the T1 built environment significantly predicted the T2 weekend MVPA (*β* = 0.24, *p* < 0.05), but the T1 weekend MVPA did not significantly predict the T2 built environment (*β* = 0.12, *p* > 0.05).

## 4. Discussion

This study found that the T1 population density, building density, street connectivity, per capita road length, mixed land utilization, number of traffic stations, distance to traffic station, distance to fitness facility, and distance to commercial area were significantly different from the corresponding T2 variables. ANOVA found that the main effects of time (from September 2017 to September 2019) on all of the built environment variables were significant, and the T2 built environment was significantly better than the T1 built environment.

The main reason may be related to Shanghai’s overall urban planning and construction goals in recent years. According to the “Practical Things Closely Related to People’s Lives that Were Completed by the Municipal Government in 2017”, 200 km of greenways were built, 400,000 m^2^ of three-dimensional greening were added, 65 civic stadiums were newly built or rebuilt, 50 fitness trails were newly built, and 210 Yizhi fitness parks were newly built or rebuilt by the end of December 2017 [21]. The “Shanghai General Planning (2017–2035)” report stated that “to optimize the urban spatial layout and create a good living environment, it is necessary to coordinate the arrangements for public service facilities such as education, culture, sports, medical care, and elderly care that are related to the vital interests of the people, and improve the quality of life service industry, the construction of high-quality, humanized public spaces, and the establishment of a community service circle that is suitable for living, working, learning, and traveling”. Relatedly, in 2018, Shanghai added a further 224 km of greenways and 400,000 m^2^ of three-dimensional greening, and the per capita green space reached 8.2 m^2^ [22]. In 2019, Shanghai promoted the improvement of 50 road traffic congestion points and built or rebuilt a further 100 citizen fitness trails, 60 civic stadiums and 300 citizens’ educational fitness centers [23]. The continuous improvement of Shanghai’s urban built environment has laid a solid foundation for the realization of individuals’ fitness and overall health.

The results showed that there was difference between T1 and T2 in weekday MVPA (*p* = 0.254), but the difference in weekend MVPA was significant (*p* = 0.037). The correlation of T1 and T2 MVPA was relatively high. ANOVA results showed that the main effect of time (from T1 to T2) on weekend MVPA was significant (*F* = 0.874, *p* < 0.05). Thus, while adolescents’ weekday MVPA was relatively stable (with little change in the short term), the weekend MVPA increased to a certain extent. This suggests that the increase in weekend MVPA may be caused by the improvement of the urban built environment. Mota et al. also found that there is a positive correlation between the number of fitness facilities near residences and the level of physical activity among children and adolescents during non-school hours [6]. We found that there was no significant gender difference in MVPA among adolescents. This may be due to the personal attributes of the subjects in our study. All of the adolescents in our study were from the downtown area of Shanghai, where socioeconomic status, parents’ education, and family economic status are higher than in other neighborhoods [24].

The research found that there was a significant positive correlation between the built environment variables and adolescents’ MVPA. This result is consistent with other cross-sectional studies that showed that the better the built environment, the higher the physical activity level among adolescents [25,26,27].

The cross-lagged panel analysis showed that the relationship between the T1 built environment and the T2 weekday MVPA was not significant, which indicates that the built environment does not significantly predict adolescents’ weekday MVPA. This may be because adolescents’ weekday MVPA mostly involves physical exercise at school. Studies have reported the important effects of school-level factors on children’s physical activity [28,29,30,31]. The cross-lagged panel analysis also found that the T1 built environment positively predicted the T2 weekend MVPA, though the T1 weekend MVPA had no significant effect on the T2 built environment. Thus, regarding the built environment and weekend MVPA, the built environment was the independent variable/predictor, while the weekend MVPA was the outcome variable. In summary, the built environment can significantly predict and affect adolescents’ weekend MVPA.

The results of this study at the theoretical and empirical levels are consistent with previous studies. Regarding the theoretical level, Sallis et al. proposed an ecological model of physical activity based on the theoretical model of social ecology, pointing out that the built environment is an important supportive environment for promoting physical activity [32]. Joshu et al. also found that the community environment was the second-most important factor (after individual characteristics) affecting the health of residents [33]. Therefore, the improvement of the built environment could play a positive role in promoting public health.

Considering the empirical level, the contribution coefficient of population density (an observable indicator in the built environment density category) in our study was 0.86, which was higher than the contribution coefficient of building density (0.72). Consistent with the results of this study, Frank et al. found that the proportion of children who walked was highly correlated with the population density in residential areas [34]. He et al. also found that community population density and intersection density were positively correlated with walking time [35]. The contribution coefficient of mixed land utilization (another important indicator when assessing the urban built environment) in our study was 0.71. Ewing et al. found that there is a strong relationship between adolescents’ physical activity and the mixed use of land [36]. The contribution coefficient of street connectivity (an observable indicator of the built environment’s accessibility) in our study was 0.81. Previous studies also found that the use of the high-density compact land development model can promote traffic-related physical activities such as walking and cycling among children and adolescents [15,37]. This model takes into account the building density, number of traffic stations, distance to traffic stations, distance to fitness facilities, and distance to commercial areas, and thus reflects the variables used in this study. The previous research confirms the practicality of the variables used in this study and the comparability of the studies.

There are limitations and challenges to the present study. First, the cross-lag design can provide information about the causal relationship between variables to a certain extent, but it lacks evidence from scientific experiments. In the future, virtual reality technology, combined with classical psychophysics methods, can be used to quantitatively manipulate various variables of the urban built environment and human behavior in virtual space, and conduct intelligent simulation experiments. Second, there are many factors affecting adolescents’ physical activity. This study only examined the relationship between urban built environments on adolescent physical activity. In the future, on the basis of extending the tracking time, more research variables (such as household income, occupation of parents) should be included to verify the stability of the relationship between the built environment and physical activity.

## 5. Conclusions

In conclusion, this study found that the urban built environment significantly affected adolescents’ weekend MVPA. The results of this study provide reference for the improvement of adolescents’ physical activity level from the perspective of urban built environment and also indicate new requirements for urban planning and construction. In the future, the density, diversity design, and accessibility of the urban built environment should be considered in urban planning in order to provide the necessary environmental foundation for the promotion of adolescents’ physical activity.

## Figures and Tables

**Figure 1 ijerph-19-08938-f001:**
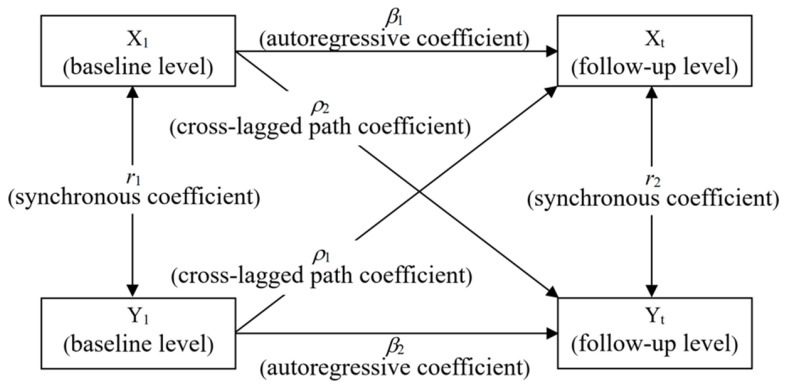
Cross-lagged panel model.

**Figure 2 ijerph-19-08938-f002:**
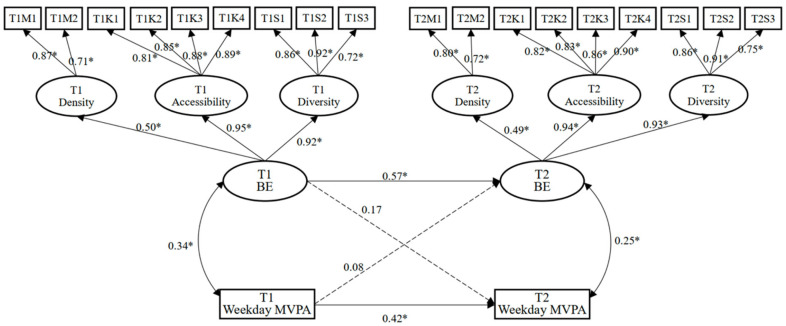
Cross-lagged panel model of built environment (BE) and weekday MVPA. Note: * *p* < 0.05. The solid line represents significant relationships and the dotted line represents non-significant relationships.

**Figure 3 ijerph-19-08938-f003:**
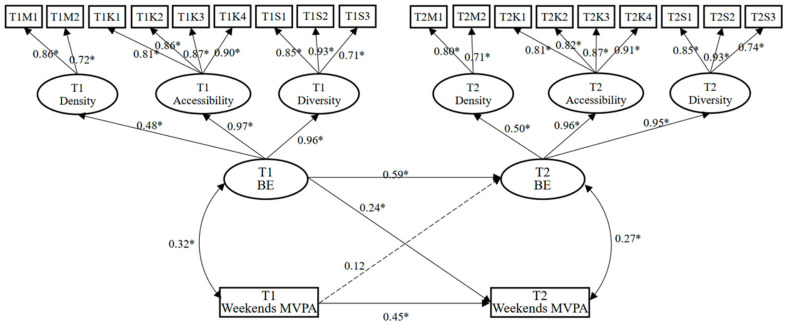
Cross-lagged panel model of built environment (BE) and weekend MVPA. Note: * *p* < 0.05. The solid line represents significant relationships and the dotted line represents non-significant relationships.

**Table 1 ijerph-19-08938-t001:** ActiGraph GT3X^+^ parameters.

	Parameter	Details
1	Monitor	ActiGraph GT3X^+^
2	Sampling frequency	30 Hz
3	Sampling interval	5 s
4	Wearing time	≥480 min/24 h
5	Number of days of data collection required for valid data collection	≥3 days on weekdays and one day on weekends
6	Physical activity intensity levels	Low: 100–1679 counts per minute (CPM)
		Moderate-to-vigorous: 1680–3368 CPM
		High: >3368 CPM

**Table 2 ijerph-19-08938-t002:** Descriptive statistics of variables.

Variables			T1		T2	
			Mean	SD	Mean	SD
Urban built environment	Density	Population density (number of people/km^2^)	52,346	20,761	55,748	21,543
		Building density (%)	0.50	0.20	0.52	0.19
	Diversity design	Street connectivity (number/km^2^)	20.33	7.41	25.73	6.87
		Per capita road length (m)	0.21	0.52	0.27	0.44
		Mixed land utilization rate	12	5	15	4
	Accessibility	Number of traffic stations	6	2	8	3
		Distance to traffic station (m)	243.22	102.67	203.28	97.83
		Distance to fitness facility (m)	156.47	132.55	134.13	122.47
		Distance to commercial area (m)	410.89	231.52	400.73	221.59
Physical activity		Weekday MVPA (min/day)	68.18	15.83	69.19	16.47
		Weekend MVPA (min/day)	65.34	18.56	74.78	22.2

**Table 3 ijerph-19-08938-t003:** ANOVA of time, gender, and built environment variables.

Variables		Main Effect				Interaction Effect	
		Time		Gender		Time × Gender	
		*F*	*p*	*F*	*p*	*F*	*p*
Density	Population density (number of people/km^2^)	2.903	0.040 *	1.453	0.446	1.127	0.364
	Building density (%)	3.726	0.034 *	0.357	0.190	0.998	0.449
Diversity design	Street connectivity (number/km^2^)	2.394	0.021 *	1.586	0.336	1.308	0.110
	Per capita road length (m)	2.765	0.047 *	0.549	0.134	0.896	0.524
	Mixed land utilization rate	4.899	0.031 *	0.183	0.165	0.513	0.127
Accessibility	Number of traffic stations	3.093	0.024 *	0.314	0.897	0.934	0.322
	Distance to traffic station (m)	2.873	0.046 *	0.748	0.564	0.099	0.443
	Distance to fitness facility (m)	10.462	0.036 *	0.114	0.276	0.424	0.056
	Distance to commercial area (m)	3.223	0.021 *	0.552	0.326	0.780	0.238

Note: * *p* < 0.05.

**Table 4 ijerph-19-08938-t004:** ANOVA of time, gender, and physical activity variables.

Variables	Main Effect				Interaction Effect	
	Time		Gender		Time × Gender	
	*F*	*p*	*F*	*p*	*F*	*p*
Weekday MVPA	1.736	0.254	0.785	0.077	0.897	0.514
Weekend MVPA	0.874	0.037 *	1.232	0.473	1.932	0.493

Note: * *p* < 0.05.

## Data Availability

The data included in this research are available upon request.

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
