# Peer review of "Association between Urban Built Environments and Moderate-to-Vigorous Physical Activity of Adolescents: A Cross-Lagged Study in Shanghai, China"

_ijerph, 2022, doi:10.3390/ijerph19158938_

Round 1
Reviewer 1 Report
Citations are needed for statements from lines 48 to 56.
Sound scientific methods. Solid data collection and analysis.
Reviewer 2 Report
The study is interesting because it attempts to clarify the causal relationship between the built environment and youth physical activity by viewing the two-year follow-up as a quasi-experiment.
Miner revision
1. 1.Introduction
Ideally, for the quasi-experiment, the only factor that has changed significantly over the two-year period and that influences the MVPA among adolescents is the built environment. Please elaborate a bit more on the rationale and validity of the quasi-experimental design. For example, disposable income could have increased over the two-year period.
2. 5.Discussion
While the discussion is generally optimistic about the results of this study, please mention any limitations of this study and future challenges. For example, restrictions regarding the quasi-experimental design or limitations regarding the choice of indicators for the built environment could be considered.
Reviewer 3 Report
Dear authors, congratulations on the study! Despite being a subject discussed daily, few references bring this level of analysis. Even my only comment is about this, I believe that if I had to go back and study a little about "cross-lagged panel", readers can "leave" their study, and consequently don't finish reading or cite for not knowing about it. Therefore, I suggest adding a short paragraph, with a good reference about this.
